# A chromatin structure-based model accurately predicts DNA replication timing in human cells

Yevgeniy Gindin[1,2], Manuel S Valenzuela[3], Mirit I Aladjem[4], Paul S Meltzer[1,*] & Sven Bilke[1]

## Abstract

The metazoan genome is replicated in precise cell lineage-specific temporal order. However, the mechanism controlling this orchestrated process is poorly understood as no molecular mechanisms have been identified that actively regulate the firing sequence of genome replication. Here, we develop a mechanistic model of genome replication capable of predicting, with accuracy rivaling experimental repeats, observed empirical replication timing program in humans. In our model, replication is initiated in an uncoordinated (time-stochastic) manner at well-defined sites. The model contains, in addition to the choice of the genomic landmark that localizes initiation, only a single adjustable parameter of direct biological relevance: the number of replication forks. We find that DNase-hypersensitive sites are optimal and independent determinants of DNA replication initiation. We demonstrate that the DNA replication timing program in human cells is a robust emergent phenomenon that, by its very nature, does not require a regulatory mechanism determining a proper replication initiation firing sequence.

**Keywords** computational model; DNA replication timing; DNase hypersensitivity; systems analysis

**Subject Categories** Genome-Scale & Integrative Biology; DNA Replication, Repair & Recombination

**Mol Syst Biol. (2014) 10: 722**

## Introduction

In eukaryotes, DNA replication is a tightly regulated process that follows a strict temporal program (Taylor, 1960; Masai *et al*, 2010). This timing program is intimately associated with key aspects of cell biology, including cell differentiation (Hiratani *et al*, 2004, 2010; Hansen *et al*, 2010), cancer progression (Fritz *et al*, 2012; Ryba *et al*, 2012; Donley & Thayer, 2013), the 3D conformation of cellular DNA (Ryba *et al*, 2010, 2012; Moindrot *et al*, 2012), and the formation of cytogenetic aberrations (De & Michor, 2011). Whereas the genome-wide replication program in eukaryotes appears nearly deterministic, individual replication initiation events display a large degree of stochasticity (Bechhoefer & Rhind, 2012). An important step in resolving this apparent discrepancy was to recognize a formal analogy between DNA replication and nucleation in one dimension (Kolmogorov, 1937; Jun *et al*, 2005), which serves as the foundation for most of today's mathematical models of DNA replication. But while the molecular components of DNA replication modeled in this formalism are mostly conserved across the domains of life, it was found that the mechanism of recognition and regulation of initiation sites varies greatly, even between lower and higher eukaryotes (Aladjem, 2007).

Particularly amendable to modeling are extreme examples of initiation site recognition: random and well characterized. *Xenopus laevis* is a representative of random initiation site selection. Modeling efforts for this organism, which need not take into account locations of initiation sites, have helped to provide theoretical answers to the so-called random completion problem (Blow *et al*, 2001; Herrick *et al*, 2002; Yang & Bechhoefer, 2008), and the global increase in the replication initiation rate throughout the S-phase suggested as one possidble solution has later been confirmed experimentally and described as a universal feature across eukaryotic replication (Goldar *et al*, 2009). *Saccharomyces cerevisiae* occupies the other end of the initiation site recognition spectrum. It's quite well characterized and efficient replication initiation sites have helped to extract a number of parameters relevant for modeling efforts, such as the average and the variance of the firing time distribution for individual initiation sites. Based on such estimates, mathematical models were able to reproduce the global timing program found in yeast (Lygeros *et al*, 2008; De Moura *et al*, 2010; Yang *et al*, 2010), thus demonstrating how the deterministic timing program emerges from individually stochastic initiation events.

Initiation site selection in metazoan genomes lies somewhere between these two extreme cases. While here too, replication initiation occurs at discrete sites in the genome, the metazoan replicator remains relatively poorly characterized, as even the most efficient sites fire in only a fraction of cell cycles (Martin *et al*, 2011; Valenzuela *et al*, 2011). This makes it more difficult to directly observe location and amplitudes of initiation (Martin *et al*, 2011; Besnard *et al*, 2012) or to extract this information from replication timing

1   Genetics Branch Center for Cancer Research, Bethesda, MD USA
2   Graduate Program in Bioinformatics, Boston University, Boston, MA, USA
3   Department of Biochemistry and Cancer Biology, School of Medicine, Meharry Medical College, Nashville, TN, USA
4   Laboratory of Molecular Pharmacology, National Cancer Institute, Bethesda, MD, USA
    *Corresponding author. Tel: +1 301 496 5266; Fax: +1 301 402 3241; E-mail: pmeltzer@mail.nih.gov

data (Baker *et al*, 2012b), contributing to the dearth of timing models for metazoan cells. Beyond these technical difficulties of obtaining a comprehensive set of robust parameters, a model built around tuning a large number of parameters (at least one for each of the 100,000 estimated initiation sites (Pope *et al*, 2013) in human cells) would remain somewhat unsatisfactory. It would also sidestep the question of what factors determine replication timing and could therefore not explain timing plasticity. Moreover, parameters for such a model would have to be re-determined for every cell state. To address these challenges, we built a minimal model and identified a genomic marker that can be utilized to predict, rather than reproduce, genome-scale DNA replication timing profiles at high resolution with an accuracy (Pearson's $r = 0.92$) rivaling that of experimental repeats ($r = 0.94$) performed in different laboratories. We use our model to demonstrate that the replication timing program can be explained by the approximate location of initiation sites alone, regardless of other factors such as exact initiation probabilities, and that initiation sites are optimally localized by DNase-hypersensitive (HS) sites.

## Results

### Mechanistic model of DNA replication

The focus of this study was to understand and predict the dynamic DNA replication timing program of human cells. Here, we took a reductionist modeling approach, including only essential components while omitting all features not required to model the timing program. In the resulting model (Fig 1A, Supplementary Fig S1), a number $N$ of rate-limiting factors independently select genomic locations and initiate replication (if the location has not yet been replicated) with probabilities specified for that location by an initiation probability landscape (IPLS). Thus, the probability of replication initiation at a given genomic location $x$ is the product of the probability of a rate-limiting factor selecting one of the unreplicated competent initiation sites at time $t$, the initiation probability assigned to that location by the IPLS and the number of available (unengaged) rate-limiting factors at time $t$.

Since the result of each simulation is determined by the choice of the input IPLS, the biological question of what determines the DNA replication timing program can be addressed by identifying the IPLS that most accurately predicts experimentally observed data. Here, human replication timing data published in (Hansen *et al*, 2010) and (Ryba *et al*, 2012) were used for this benchmark. Both datasets

report the average behavior of cell populations. We compared our model's prediction, averaged over millions of Monte Carlo-simulated cell cycles, to these benchmark datasets. The concordance between predictions generated by the optimal model (see below) and the experimental data is striking at the 500-bp resolution used in our simulations (Fig 1B and Supplementary Fig S2), recapitulating peaks and valleys of replication timing on a chromosome-wide scale (Fig 1C).

### Predictive power of static genomic features

Earlier studies (Cayrou *et al*, 2011; Martin *et al*, 2011; Valenzuela *et al*, 2011) had indicated that DNA replication initiation is more likely to occur in the vicinity transcription start sites (TSSs). Thus, as a starting point, we tested the predictive capacity of an IPLS where we assigned a constant, time-independent high initiation probability to all TSSs annotated in RefSeq (Pruitt *et al*, 2005) and low probabilities everywhere else (see Supplementary Information S1 for further discussion). Despite the simplicity of these assumptions, the resulting timing prediction is quite similar (average $r = 0.69$ across four cell lines Fig 2A) to the experimental data. Testing other sequence features that were previously associated with replication initiation generates similarly good predictions: CpG islands (Meyer *et al*, 2013) ($r = 0.64$), GC content (Meyer *et al*, 2013) ($r = 0.58$), and predicted G4-quadruplexes (Besnard *et al*, 2012) ($r = 0.55$) (Fig 2A and Supplementary Fig S3). Remarkably, an IPLS based on a structural feature of the DNA molecule, namely its solvent-accessible surface (Greenbaum *et al*, 2007), produced profiles ($r = 0.51$) only slightly less predictive than some of the other, more commonly discussed factors (Fig 2A and Supplementary Fig S3). However, such invariant properties of the genome cannot account for timing plasticity observed across cell types (Hansen *et al*, 2010). We therefore hypothesized that dynamic genomic landmarks would generate models better suited to capture differentiation lineage-specific timing plasticity.

### DNase-hypersensitive sites are the main determinants of DNA replication timing

Utilizing the recently published (ENCODE Project Consortium & others, 2011; Rosenbloom *et al*, 2013) ENCODE data, we generated IPLSs from all 167 cell-specific datasets available for the cell lines in the Hansen data by assigning an initiation probability proportional to the ENCODE amplitude, simulated the timing patterns, and compared the results to the empirical DNA replication timing data for

**Figure 1. Mechanistic model overview and simulation results.**

A   Mechanistic model inputs are initiation probability landscape (IPLS) and the number $N$ of replication forks. The DNA replication program is executed on a simulated cell population (a single cell is depicted). Simulated cells can be either in a non-replicating state (denoted as "G") or in a replicating state ("S"). At the start of the simulation, all cells are in the G state. Transition from G to S occurs randomly. When in the S state, free (red) rate-limiting forks select a random location and bind with a probability set by the IPLS or remain unengaged otherwise. Once engaged (green), replication occurs bi-directionally until forks collide returning to their unengaged state, restarting the process until the genome is replicated. The model periodically queries each cell's replication progress. Once the genome is replicated, the cell enters G state, repeating the process until simulation is terminated.

B   Simulated and empirical DNA replication timing are highly correlated. Each point in the contour plot represents a replication time assignment for a 500-nt bin on chromosome 12 of GM06990 cells. Simulated replication timing assignment is given on the *y*-axis, and the experimentally derived assignment is given on the *x*-axis. Contour lines are meant to aid in interpretation. *R*-value represents Pearson's correlation between simulated and empirical data.

C   Simulation based on DNase-HS sites produces high-fidelity replication timing predictions. The simulated timing program (red) generally lies on or between two experimental datasets plotted on the same axes, the Hansen (Han) dataset (red) and the Ryba (Ryb) dataset (blue). The density of DNase-HS sites (DNase I) is plotted with higher-density regions colored darker. The stated correlation *R*-values and all the data that are shown are specific to chromosome 14.

    

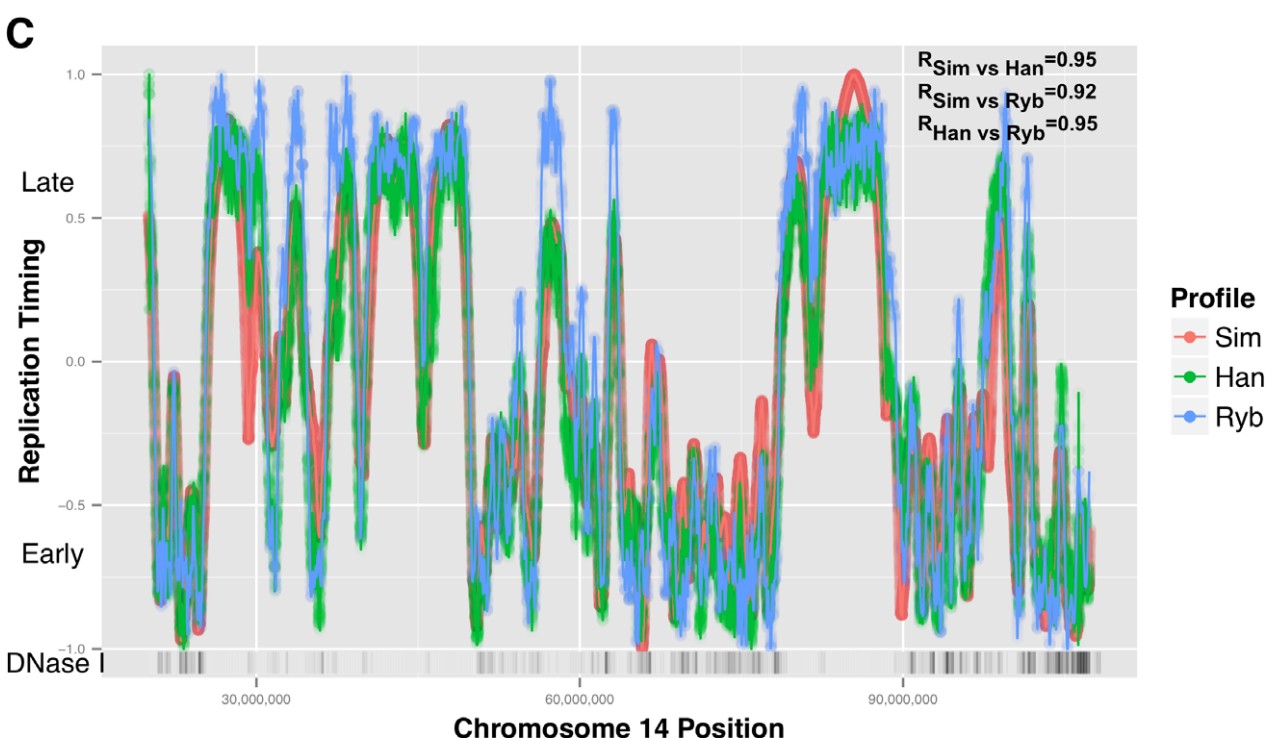

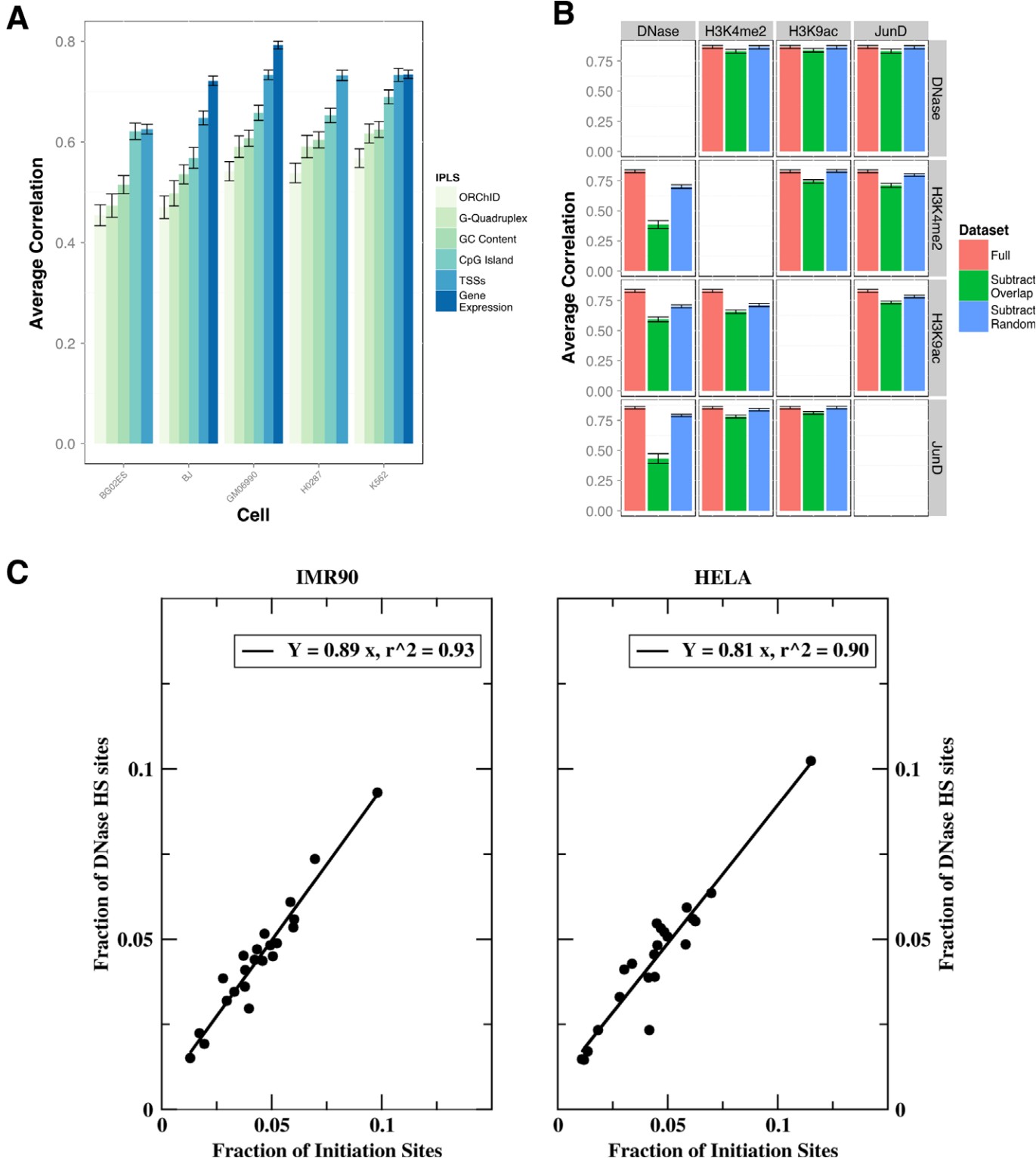

corresponding cells (Supplementary Table S1). Nearly one-half (77 out of 167) of the probed ENCODE marks produce better predictive models compared to the best (TSS-based) static model (Table 1 and Supplementary Fig S4). Notably, the gene expression-based model (AffyExonArray, $r = 0.75$) did not show a measurably improved accuracy in comparison with the static TSS model ($r = 0.69$). The top-ranking model ($r = 0.87$) is based on an IPLS derived from DNase-HS sites. This is followed by models derived from activating chromatin marks such as H3k9ac ($r = 0.83$), H3k4me2 ($r = 0.83$), or transcription factor binding (e.g., JunD $r = 0.86$).

**Figure 2.  DNase-HS sites are main independent determinants of DNA replication timing.**

A  Simulations based on genome sequence features (GC content, CpG islands), or local genome conformation (ORChID, G-quadruplex), RefSeq annotated transcription start sites (TSS) and gene expression levels (where available) in five cell lines. Shown is the correlation with the Hansen (Han) dataset averaged over 22 autosomal chromosomes, and error bars represent the standard error of the mean.

B  Mutual independence of representative top-ranking ENCODE marks (DNase, JunD, H3K4me2, H3K9ac) is probed by eliminating co-localized genomic marks in pairwise comparisons. The results of these 4 (datasets) × 3 (overlaps) = 12 sets of simulations are presented in a 4 × 4 matrix format: rows indicate the dataset that was used to generate the initiation probability landscape (IPLS), and columns indicate the subtracted dataset. Each panel plots the correlation to the experimental timing data in K652 cells (the only set for which all annotations were available) for the full dataset (red), the non-co-localized marks (green), and a "random" dataset (blue) from which the same number of (not necessarily overlapping) marks was removed. Error bars represent standard error of the mean.

C  The number of initiation sites had been shown earlier to be non-trivially distributed across chromosomes (Besnard *et al*, 2012). Comparison of the number of DNase-HS sites in IMR90 and HELA with the number of initiation sites on each chromosome reveals a tight correlation between the two. Each data point in the plot represents the fraction (sum = 1) of initiation and DNase-HS sites, respectively, on a autosomal chromosome (see also Supplementary Fig S6).

**Table 1.  Top DNA replication timing predicting initiation probability landscape (IPLS) sources**

| IPLS source | Average correlation |
| --- | --- |
| DnaseDgf | 0.865 |
| CCNT2 | 0.855 |
| JunD | 0.855 |
| FaireSeq | 0.854 |
| ZNF384 | 0.849 |
| COREST | 0.849 |
| CEBPB | 0.847 |
| MAZ | 0.842 |
| TBLR1 | 0.839 |
| eGFP-JunD | 0.835 |
| ZNF-MIZD-CP1 | 0.834 |
| H3K9acB | 0.829 |
| H3K4me2 | 0.829 |
| HCFC1 | 0.828 |
| UBTF | 0.828 |
| HMGN3 | 0.828 |
| BHLHE40 | 0.827 |
| TBP | 0.827 |
| DnaseSeq | 0.825 |
| H3K4me1 | 0.824 |

We hypothesized that the ability of more than one epigenetic mark to predict DNA replication timing with high fidelity is a consequence of the fact that many chromatin marks tend to co-localize (Thurman *et al*, 2012) and that, in isolation, some marks would lose much of their predictive value. To test this possibility, we performed simulations based on reduced sets, where mutually co-localized marks were removed (Fig 2B and Supplementary Fig S5). Remarkably, among the tested top-ranking genomic marks selected for this analysis (histone H3K4me2, H3K9ac, transcription factor JunD, and DNase-HS sites), only DNase-HS sites fully retained their ability to predict replication timing in all pairwise comparisons. For all other marks, accuracy of the timing prediction was substantially reduced when removing overlaps with DNase-HS sites, even when accounting for the reduced set size. We further explored whether these same marks co-localize with empirically determined DNA replication initiation sites (Besnard *et al*, 2012). Our results show that JunD, H3K4me2, and H3K9ac sites overlap DNA replication origins

only so long as they also overlap DNase-HS sites (Supplementary Fig S6). We therefore conclude, based on the available data, that DNase HS is the main independent determinant of replication timing. This conclusion is further supported by observing that almost half of the DNase-HS sites in HeLa (47%, *P* < 1E-6, OR=5.0) and IMR90 (47%, *P* < 1E-6, OR=3.8) cells are located within 500 bases of empirically determined initiation sites (Besnard *et al*, 2012). Also, the non-trivial distribution of initiation sites across chromosomes, with the density of initiation sites varying substantially between chromosomes, is closely recapitulated by DNase-HS sites (Fig 2C and Supplementary Fig S7).

**DNA replication timing plasticity across cell lineages and species and its alteration as a result of chromosomal fusions**

Replication timing shows remarkable plasticity across differential lineages (Donley & Thayer, 2013) and in cancer cells (Ryba *et al*, 2012). Utilizing DNase-HS data for three cell lines (BJ, GM06990, K562), for which matching experimental timing and DNase-HS data were available, we performed DNA replication simulations and hierarchical clustering of the simulated and experimental data (Fig 3A). The model predictions tightly cluster with the experimental data for the matching cell and also recapitulate the closer relatedness of GM06990 and K562 cells (both of hematopoietic origin) in comparison with BJ (fibroblast). Using stringent parameters (see Materials and Methods), we identified 60 genes (Supplementary Table S2) in regions with replication timing variable regions between GM06990 and K562 cells and found a significant enrichment for interferon and hemoglobin complexes (DAVID (Huang *et al*, 2009) *P*-value 3.3E-12 and 2.3E-10, respectively), including the human β-globin locus (Fig 3B)—in line with phenotypic properties of these cells.

The accuracy of our model predictions in human cells suggested that the same mechanism will likely work in other mammalian cells. Currently, the lack of simultaneous availability of both replication timing and DNase-HS data for the same cells limits the ability for a broader analysis. To test the applicability of our model to mouse embryonic fibroblast cells, we compared replication timing predictions generated from DNase-HS sites in NIH/3T3 cells (ENCODE Project Consortium & others, 2011) to observed timing data in mouse embryonic fibroblast cells (Hiratani *et al*, 2010). The average Pearson's correlation between model prediction and experimental replication data is 0.85 (Supplementary Fig S8), confirming that our model can be extended to other metazoan cells.

Recurrent chromosomal fusions are found in many cancers (Rowley, 1973; Delattre *et al*, 1992; Tomlins *et al*, 2005). In acute lymphoblastic leukemia, the well-characterized t(12;21)(p13;q22);

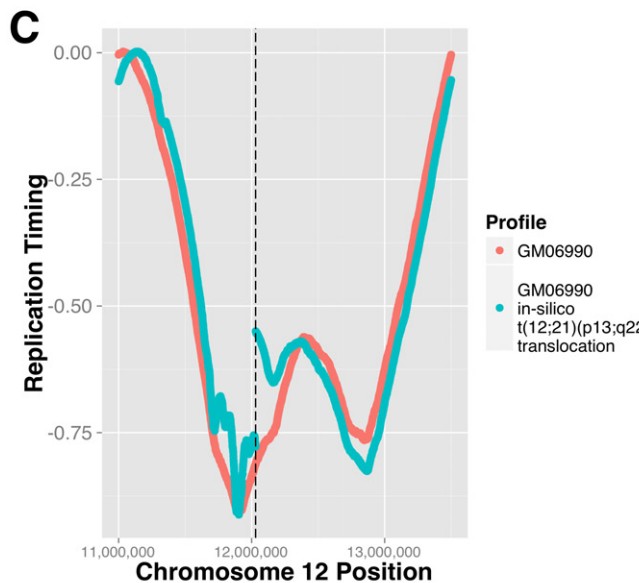

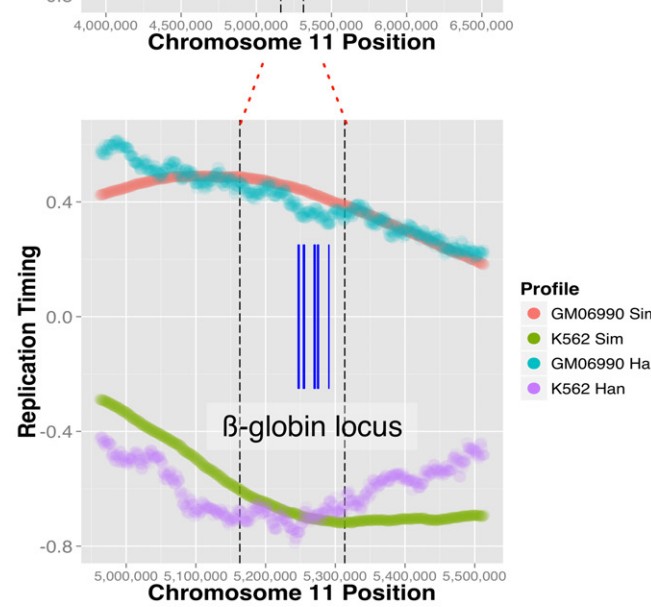

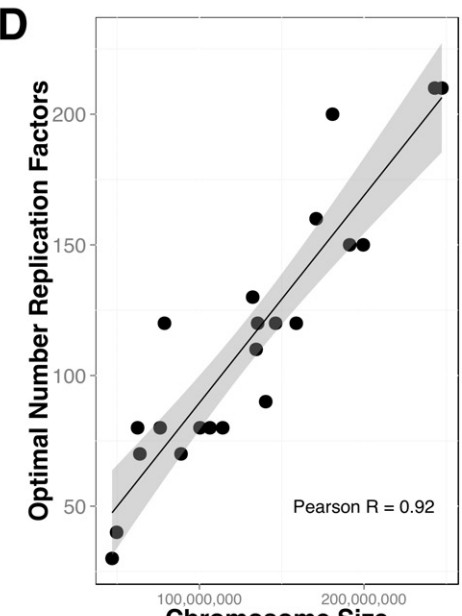

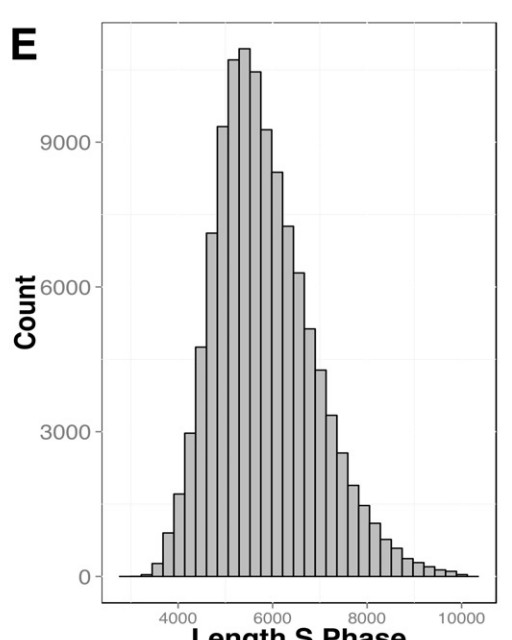

**Figure 3.  Mechanistic model is highly reflective of the underlying DNA replication timing biology.**

A  Hierarchical clustering and correlations heatmap of simulated and empirical data. Individual correlations (Pearson's) are noted in the matrix for every dataset pair. Simulations are consistently placed closest to the associated experimental data. The stated correlations are based on simulations which were not optimized for flow-sorter settings.

B  Analysis of timing plasticity between simulated GM06990 and K562 cells identified, among other regions, differential timing in the β-globin locus (indicated by dashed lines, genes marked in blue). Hansen dataset (Han) is shown for reference.

C  A translocation event simulated *in silico* in GM06990 cells qualitatively reproduces the timing discontinuity observed (Wiemels *et al*, 2000) at a *TEL-AML1* translocation in ALL. Replication profile of translocated (blue line) and normal (red line) are shown on the same genomic axis, and the dashed line signifies the translocation coordinate.

D  DNA replication timing profiles were simulated using initiation probability landscape (IPLSs) derived from GM06990 DNase-HS data, noting the number of replication factors that produced highest correlation for each chromosome. Solid line represents a linear fit (shading area denotes the 95% confidence interval). The linear regression curve estimates that the number of forks per megabase is given by $N = 10.24 + 7.9E\text{-}7 * x$, where $x$ is chromosome length. The Pearson's correlation between the optimal number of replication forks and chromosome length is 0.92.

E  Histogram illustrating the distribution of the lengths of the S-phase in a simulated asynchronous cycling population of GM06990 cells.

*ETV6-RUNX1* fusion is accompanied by an abrupt change in DNA replication timing near the fusion site (Ryba *et al*, 2012). Our model reproduces this behavior when inducing (see Supplementary Information S1) an *in silico* t(12;21)(p13;q22) translocation in GM06990 lymphoblastoid cells (Fig 3C). This behavior is also reproduced when comparing replication timing at the *in silico*-induced breakpoint in GM06990 cells with observed replication timing in REH cells, which harbor the translocation (Supplementary Fig S9). The results show that replication timing is not determined at the site of the breakpoint. Instead, the timing pattern arises from the combined influence of the DNase-HS sites situated on either side of the break. The discontinuity, observed experimentally and reproduced in the simulation, is the result of mapping physical coordinates of the rearranged chromosome 12 onto the normal genome.

**Modeling parameters**

The proposed model has remarkably few parameters. In addition to an IPLS and an optional technical variable (see below), there is only one adjustable parameter, namely the maximum number ($N$) of replication forks that can be active simultaneously. As $N$ is not set *a priori* (Supplementary Fig S10), we performed a series of simulations identifying, for each chromosome, the optimal $N$ that generates the closest match to the experimental data. We find that the optimal $N$ grows linearly with chromosome length at a rate of 1 fork per 1.3 mega bases (Fig 3D), compatible with the assumption that the stochastic process governing replication does not substantially differ between chromosomes. Subsequently, we used the estimate from the linear regression curve in this experiment for $N$. With this setting, the predicted median length of the S-phase is 5,965 (mean = 6,134) simulation steps (Fig 3E) in GM06990 cells. In real human cells, replication forks move at a speed of about 50 bases per second (Alberts *et al*, 2008). With a 500-bp model resolution (and two forks moving in opposite directions in each simulation step), the predicted median wall-clock time for the S-phase is $t = 5965$ steps * 500 bases/(50 bases/s)/(2 step) = 8.3 h, in line with the experimentally observed duration of 6–10 h. The optional technical variable mentioned previously governs the simulation of a flow-sorter, which in laboratory experiments divides asynchronously replicating cells into S-phase fractions (Ryba *et al*, 2011). As the actual gate settings used were not available (Hansen *et al*, 2010), numerical optimizations (see Supplementary Information S1) were used, further improving the similarity of the best (DNase HS-based) simulated models from $r = 0.89$ (with 5 equidistant gates in GM06990

cells) to $r = 0.92$, a level approaching the limit of experimental noise ($r = 0.94$ between experiments performed in different laboratories).

**DNA replication timing is highly robust**

With only a single adjustable biological parameter, and thus no real risk of over-fitting the data, the accuracy of our model predictions is exceptionally high, suggesting a high degree of robustness of the proposed model. One potential limitation is the completeness of genomic annotations. To test its importance, we built a series of models by randomly sub-sampling DNase-HS annotations. The predictions were essentially unchanged despite removing up to 75% of DNase-HS sites with the accuracy degrading gradually beyond this point (Supplementary Fig S11A). We conclude that the local replication timing program emerges from the collective contribution of adjacent initiation sites, and as a systems phenomenon, it is largely independent from individual sites.

The model also shows a large degree of insensitivity with respect to specific modeling choices. We wondered how strongly the specifics of assigning probabilities in the IPLS based on ENCODE amplitudes affects the simulation results. For the simulations presented so far, the local initiation probability was set to be proportional to the ENCODE amplitude. We tried alternate assignment functions (Supplementary Fig S11B) which resulted in only insignificant changes in the accuracy of the model prediction (linear $r = 0.86$, square $r = 0.86$, square-root $r = 0.84$) and even when assigning the same *constant* value to all sites ($r = 0.86$). On the molecular level in real cells, this implies that, once a site is competent to initiate, the probability that it is going to do so does not substantially affect the global replication timing program. We conclude that the relevant information provided by the ENCODE data, with regard to DNA replication timing, is *location* while *amplitude* is irrelevant. In our early simulations, we had included a small background initiation rate outside of the high efficiency initiation sites demarcated by DNase-HS sites. This choice, too, was found to not affect the accuracy of the model prediction (see Supplementary Information S1), even when assigning a zero background initiation rate, that is, when initiation exclusively occurred at high efficiency sites (Supplementary Fig S12).

# Discussion

Here, we present a mechanistic model that fully predicts replication timing in human cells, without the need to adjust any parameters

for new cell types. Once the number of rate-limiting factors and the genomic landmark that optimally generates the IPLS had been identified, the same constant choices produced accurate timing predictions for any tested cell type. Designed in a reductionist spirit, we attempted to omit all details from the model that are not required to understand the timing program (Supplementary Fig S1A and B). We wondered if the replication fork collision mechanism in the model, which dynamically determines the distance a fork travels, could be removed by instead using the density of DNase-HS sites in the vicinity (see Supplementary Information S1) to assign a replication time. All such models produced substantially worse predictions (Supplementary Fig S13 and Supplementary Information S1), indicating that the collision mechanism is a required aspect of the model. Therefore, while genomic regions dense in DNase-HS sites delineate early DNA replication regions (Fig 1C), they are not sufficient, on their own, to predict DNA replication timing.

An explicit separation of replication into licensing and initiation steps proved also to be unnecessary. This separation is known to be an essential molecular mechanism to avoid over-replication: licensing occurs exclusively in late M/early G1 by assembly of the so-called pre-replication complex (PreRC) at potential initiation sites, with initiation occurring later in the S-phase by conversion of the PreRC into bi-directional replication forks through phosphorylation and recruitment of other factors (Machida *et al*, 2005). In our model, the IPLS subsumes these two steps (over-replication itself is prevented by explicitly keeping track of replicated regions); the initiation probability at a given site represents the product of the biological probabilities to first assemble and later activate the PreRC. The above described intrinsic robustness of the model with respect to the assignment of probabilities in the IPLS is remarkable in this context. It implies that the factor dominating the timing program is the selection of the *location* of PreRC assemblies. Our model predicts (Supplementary Fig S11B) that the empirical timing pattern will emerge even if all PreRCs, once assembled, have the same constant probability of being subsequently activated unless the site is passively replicated. While, to our knowledge, this possibility has not been tested in metazoan cells, it has broadly been shown to be the case in yeast (Yang *et al*, 2010), where a majority of initiation sites were demonstrated to have a "potential initiation efficiency," with the initiation probability remaining larger than 0.9 after correcting for passive replication.

Remarkably, we were not required to introduce a time-dependent IPLS in order to precisely predict the global replication timing pattern. Earlier models (Hyrien & Goldar, 2010; Yang *et al*, 2010) used location and explicit time-dependent initiation rates $I(x, t)$ to force individual initiation sites to fire, on average, at the right time to reproduce the global timing pattern. While these approaches elegantly reconcile the orchestrated global replication timing program with the stochastic nature of individual initiation events, they do not ultimately address what determines the local initiation timing. Instead, they reproduce, but do not predict, replication timing. This is because these models rely on existing timing data, for each cell type, in order to fit a large number of variables, one or more for each initiation site. In contrast, in the model presented here, the global timing program results from the spatial distribution of initiation sites, and a determination of individual firing rates was therefore not necessary. Once the genomic landmark that optimally locates initiation sites, DNase HS, was determined, timing could be predicted

for all cell types. We expect that the basic mechanism described here will also work in other metazoan cells. Indeed, we found an excellent agreement between our model prediction and experimental timing data in mouse embryonic fibroblast cells (Supplementary Fig S8).

Another important reason to use a time-dependent, globally increasing initiation rate throughout S-phase in earlier models is to stabilize the S-phase length, thus avoiding the random completion problem (Blow *et al*, 2001; Herrick *et al*, 2002; Yang & Bechhoefer, 2008). These predictions were confirmed by a recent analysis (Goldar *et al*, 2009), uncovering a universal behavior of the global initiation firing rate across a number of species. How does this reconcile with the time-independent IPLS presented here? The firing rate in our model depends not only on the explicitly time-independent IPLS, but also on the number of unengaged rate-limiting factors, which dynamically changes over time, as well as on the search time to find unreplicated origins, which differs between early and late replicating regions as a result of the difference in the density of initiation sites. A numerical analysis of the global initiation rate (Supplementary Fig S15) shows a remarkable qualitative similarity to the universal patterns described in Goldar *et al*, (2009). It will be interesting to see whether it is necessary to extend our model by including a more detailed replication factor diffusion process, such as the sub-diffusive model discussed in Gauthier & Bechhoefer (2009), in order to obtain a quantitative match with experimentally determined global initiation rates in human cells.

We identified DNase hypersensitivity as the optimal IPLS predicting the DNA replication timing in metazoan cells. This suggests that DNA replication timing is largely determined mechanistically: locally by DNA accessibility as the dominant factor modulating the likelihood of forming competent initiation complexes and globally by the process of colliding replication forks—a reduced representation of the known molecular processes. This interpretation implies a causal relationship, where the distribution of accessible genome regions determines DNA replication timing. Recently, a tight correlation, although significantly weaker (Pearson's $r = 0.8$) compared to the best models tested here, between replication timing and the first eigenvector of the HiC contact probability matrix has been reported (Ryba *et al*, 2010), suggesting that the 3D genome organization may play a prominent role in DNA replication timing, for example, via replication factories or by determining the boundaries of replication domains (Baker *et al*, 2012a). It may therefore seem surprising that our accurately predictive model does not require any reference to the spatial genomic organization. It could be that both phenomena, the distribution of DNase HS and 3D conformation, have a common cause. Yet, it is generally believed that DNase-HS sites are established by transcription factors dislocating and/or limiting the movement of histones (Felsenfeld *et al*, 1996). It therefore seems reasonable to speculate that the distribution of DNase-HS sites might itself contribute to the control of the genomic conformation.

In summary, provided with a proper "IPLS"—a mathematical construct that encodes the location information, the model predicts the replication timing program and recapitulates cell-specific timing patterns, including abnormal timing behavior in cancer cells. These results strongly support the concept that replication timing is a stochastic process ultimately determined by chromatin structure, which itself is a consequence of the topological organization of genes and functional regulatory elements on the chromosome as encoded in the DNA sequence.

# Materials and Methods

## Software implementation

The custom-written software (Replicon) is capable to simulating genome replication and recording various associated measurements, such as DNA replication timing. Replicon is written in C++ and can be executed in a multi-threaded mode. In our experiments, a typical simulation of a human genome-wide DNA replication profile took about 15 min when executed in parallel: 22 simulations each running on a 4-core, 2.93-GHz Linux node.

Replicon source code is included in the accompanying supplement as is a set of instructions on how to generate IPLS files from BED-formatted files. Also included with the supplement is the set of IPLS files (based on DNase I-HS site data) and replication timing predictions for GM06990 cells.

## Simulated replication time assignment to genome coordinates

The simulation consists of millions of simulated cell cycles. The assignment of replication time to genome coordinates starts by first separating the cell population, according to each cell's DNA content, into one of six bins (akin to a flow-sorter sort). The replication time is calculated for each genome coordinate (500-bp resolution) by taking the average of the product of the bin number (1 through 6) and the number of times the genome coordinate in question was observed in each bin.

## Flow-sorter gating optimization

We used a simulated annealing algorithm to approximate DNA flow-sorter bin boundaries with the objective to minimize the Euclidean distance between simulated and experimentally derived DNA replication timing profile. Starting from a state where flow-sorter bin boundaries were randomized, replication timing was simulated based on DNase DGF data for GM06990 cells. The neighboring state was calculated by perturbing a randomly chosen bin boundary. The new boundary value was chosen from normal distribution, where $\mu$ was set to the old boundary and $\sigma$ to a value of 1.

## IPLS generation

Utilizing a 500-bp resolution, the probability of initiating replication at any given genomic location was set either to a scaled value of an attribute of interest or to a background frequency of 1E-4, whichever was greater. Scaling was achieved using the formula $x/\max(x)$, where $x$ is the attribute of interest. All DNA replication initiation landscapes, unless stated otherwise, were generated from a local copy of the UCSC ENCODE database (Rosenbloom *et al*, 2013), where the data attribute "score" was used as the attribute of interest. For GC-content IPLS, the probability of DNA replication initiation was scaled to the "sumData" attribute of the "gc5Base" annotation table. For CpG island IPLS, the probability of DNA replication initiation was scaled to the "obsExp" attribute of the "cpgIslandExt" annotation table. For DNA G-quadruplex (G4) IPLS, the probability of DNA replication initiation was scaled to the length of the G4 motif. The G4 motifs were identified using a regular expression as described in Todd *et al*, (2005). ORChID IPLS was based on "wgEncodeBuOrchidV1.bigWig" annotation file available at the UCSC Genome Browser (http://genome.ucsc.edu/), where the intensities of hydroxyl radical accessibility were averaged over non-overlapping 500-bp windows. The transcription start site (TSS) IPLS was set to a constant probability of 1.0 for every genomic region annotated as "txStart" in the "refGene" table.

## Generation of reduced-model IPLSs

For each set of genome annotations in a pairwise comparison, we identified and removed co-localized genome regions, generating the "Subtract Overlap" reduced model for each model in the comparison. The "Subtract Random" model was generated by removing randomly chosen genome regions from each model in the comparison, such that the number of regions in "Subtract Overlap" and "Subtract Random" models was equal.

## *In silico* ETV6-RUNX1 translocation

We generated t(12;21)(p13;q22) chromosomal translocation *in silico* by joining GM06990 DNase DGF data for chromosomes 12 and 21—producing an *ETV6*-RUNX fusion gene using molecularly mapped breakpoint coordinates (Wiemels *et al*, 2000). We then simulated replication timing for two fused chromosome products and compared simulated replication timing data for translocated and un-translocated chromosome 12.

## Robustness

The effect of deleting DNase-HS sites was investigated using DNase DGF data available for GM06990 cells. At each iteration of the algorithm, we erased an ever-increasing fraction of DNase sites and generated a corresponding DNA replication initiation landscape.

## DNA replication plasticity regions

DNA replication plasticity regions were identified using custom-developed software. First, a DNA replication difference profile was derived for a given pair of DNA replication timing profiles by subtracting one DNA replication profile from another for matching genome coordinates. The distribution of differences was observed to follow normal distribution. Using the normal distribution, a *P*-value was assigned to every 500-bp non-overlapping genome bin (the resolution of our model) in the difference profile. A DNA replication plasticity region was identified as such if at least three consecutive bins were assigned a *P*-value of 0.001 or less.

**Supplementary information** for this article is available online: http://msb.embopress.org

## Acknowledgments

This research was supported by the Intramural Research Program of the NIH, National Cancer Institute, Center for Cancer Research. MSV was supported by NIH grant SC1CA1 138180. We thank Sean Davis, Kevin Gardner, and Subhajyoti De for discussions.

## Author contributions

YG and SB designed and performed statistical analyses and interpreted the data, wrote the manuscript, and designed experiments with significant contributions made by MSV, MIA, and PSM. YG developed optimization routines. SB conceived the proposed model and implemented the simulator. SB and PSM conceived the study and supervised the work.

## Conflict of interest

The authors declare that they have no conflict of interest.

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
