## [Review Process File · Molecular Systems Biology]

A chromatin structure based model accurately predicts DNA replication timing in human cells

Yevgeniy Gindin, Manuel S. Valenzuela, Mirit I. Aladjem, Paul S. Meltzer, Sven Bilke

Corresponding author: Paul S. Meltzer, National Cancer Institute

Review timeline:	Submission date:	17 September 2013
	Editorial Decision:	17 October 2013
	Revision received:	13 January 2014
	Editorial Decision:	29 January 2014
	Revision received:	30 January 2014
	Editorial Decision:	05 February 2014
	Revision received:	11 February 2014
	Accepted:	12 February 2014

Editor: Maria Polychronidou

Transaction Report:

1st Editorial Decision

17 October 2013

Thank you again for submitting your work to Molecular Systems Biology. We have now heard back from the three referees who agreed to evaluate your manuscript. As you will see from the reports below, while reviewers #1 and #2 are cautiously supportive, reviewer #3 raises significant concerns, which should be convincingly addressed in a revision of the manuscript.

We would particularly like to draw your attention to a serious concern that was raised by reviewer #3 and refers to the previously published study of Baker et al., (PLoS Computational Biology, 2012), which concludes that the causal determinant of DNA replication timing is 3D structure, while DNase I hypersensitive sites play an incidental role. As this point could potentially severely undermine the main conclusions of the manuscript, it will be essential to convincingly demonstrate the causal role of DNase I hypersensitivity in replication timing and rule-out alternative mechanisms.

Without repeating all the points listed below, other fundamental issues raised by the referees are the following:

- The presented results/conclusions need to be correctly placed in the context of existing literature and previously proposed alternative interpretations.
- Referee #3 has raised a series of concerns regarding the employed methodology and the related assumptions/parameters.

If you feel you can satisfactorily deal with these points and those listed by the referees, you may wish to submit a revised version of your manuscript. Please attach a covering letter giving details of the way in which you have handled each of the points raised by the referees. A revised manuscript

will be once again subject to review and you probably understand that we can give you no guarantee at this stage that the eventual outcome will be favorable.

REFeree REPORTS:

Reviewer #1:

In this paper, the authors used Monte Carlo simulations to predict replication timing only 2 parameters: IPLS (initiation probability landscape) and N, the number of simultaneously active replication forks. IPLS is adopted from various genomic and epigenomic features, which basically determines the location of replication initiation. By using this prediction model, they show that one single feature - DNase hypersensitivity - can completely account for the timing profile. Other epigenetic marks correlate with RT due to their co-variation with DNase HS sites. The remarkable robustness of the model to random removal of 3/4 of DHSs is exactly what is expected of a replication system, given what we know of replication regulation in mammalian cells, and provides additional confidence in this model. The fact that it is independent of probability at each site challenges one of the major models of replication timing regulation; these data suggest it is the locations of sites not so much their efficiency. While readers may not be prepared to conclude that every DHS is an origin or vice versa, and I suspect many replication people will be resistant to this paper, the point here is that the simple principles of this model can account for the patterns of replication seen. This is a provocative and important study and should be published. There is just one point that I feel needs to be addressed and for which I would like to see the response. The discussion of the Besnard origins is very confusing:

"A similar picture emerged when integrating experimental replication initiation data (Besnard et al, 2012), where initiation activity is strongly diminished at non-DNase HS overlapping marks when compared to the full dataset (Figure S6)." Do the origins look the same or better or worse than DNaseI alone?

If you take the origins and subtract DNaseI, what does that model look like?
How do you interpret these results (depending on what they are)?

Other points (it was difficult to point to these things without page numbers)

1) Some of the references are not correct in the second sentence of the introduction:

a) It is intimately associated with key aspects of cell biology, including cell differentiation (Hansen et al, 2010; Ryba et al, 2011b), - the first paper to show replication timing changes during differentiation at all is Hiratani et. al., PNAS, 2004, then the first to show it by genome-wide methods is Hiratani et. al., PLoS Biology 2008.

b) cancer progression (Donley & Thayer, 2013; Fritz et al, 2012), - this is where Ryba et. al. Genome Research 2012 should be referenced

c) the 3D conformation of cellular DNA (Ryba et al, 2012; Moindrot et al, 2012) - the first paper to show this is Ryba et. al. Genome Research 2010.

2) Some statements should be toned down:

- a) Bottom of first intro page: "robustly determined" - I suggest changing these phrases to: "can be accounted for"
- b) Middle of pdf page 6 - same thing "main independent determinant".

3) The correlations to cell types using cell type specific DHS data look good, but I did not see how the DNaseI model for one cell type matches or deviates from the replication timing of a different cell type. A simple correlation matrix might be nice to show.

4) Figure 3C - this figure is not entirely clear. First, it is confusing because if this is an in silico translocation, and if DNaseI follows replication timing, shouldn't this be obvious. Clearly they do

not have DHSs in the translocated cell line but, if the authors wish to compare their data to actual data, at a minimum they should use the replication timing profiles from a translocated cell line, which are published and available.

5) There was a paper by Eric Bouhassira in *Genome Research*, 2009 where he predicted replication timing from the promoters of transcribed genes. How do the authors' data differ? Clearly the match here is better - can the author's provide a satisfactory explanation.

6) Figure 1B was not clear to me at all. What are the lines? Can the authors explain a bit more. Also, it seems that since it is showing a correlation, that some kind of R value should be shown in the figure, if I understand it correctly?

7) In Figure 3A, it would help to use correlation values in addition to the distances in the dendrogram to see the differences better.

Reviewer #2 :

Girdin et al. describe an elegantly simple computational model for human DNA replication and use it to explore the factors that determine replication timing. They come to the surprising, but apparently robust, conclusion that timing is determined almost exclusively by the location of origins and not any intrinsic timing or efficiency characteristic of the origins. Moreover, they show that the position of origins is best predicted by DNase I hypersensitive sites (DHSs) and that other genome features that correlate with origins do so primarily because they too correlate with DHSs. These results have important mechanistic implications for biochemical models of how replication is regulated. As such, it will be of interest to a broad audience interested in DNA replication and genome stability. Nonetheless, the following issues need to be addressed before publication.

The conclusion that only the location of origins is important, and that the timing or efficiency of individual origin firing is irrelevant, is initially counterintuitive. However, after reading the discussion very carefully, I think I understand what is going on. It seems to me that the key is the going to be the density of origins/DHSs, with more origins/kb specifying early regions and less origins/kb specifying late regions. If this interpretation is correct, the authors should depict origin density v. timing, perhaps as heat maps, to make this point clear. If not, they should explain, at least to me, why this interpretation is incorrect. It seems to me that this interpretation is conceptually similar to the Multiple Initiator Model of Yang et al.

<<http://www.ncbi.nlm.nih.gov/pubmed/20739926>>, which the authors reference, in which the behavior of any one initiator (MCMs for Yang, DHSs for Girdin) is uniform and inconsequential; what matters is the number of initiator (at an origin for Yang or in a region of initiation for Girdin). A key factor in this interpretation is the difference in resolution of DHS maps (~1kb) and the replication profiles (~100kb) (and here I am referring to the resolution of the experimental replication profiles and the output of the model (i.e., Figure 1C), not the 500 bp resolution at which the model is calculated), which allows many DHS sites in every region of replication initiation, even in relatively DHS-poor late replicating regions. However, it took me a lot of work to make this connection. This analysis is implicit in the description and discussion of the paper's results, but I think it would be a great service to many readers, who may not make the required conceptual connections, if it was spelled out much more explicitly. It will also make the paper more likely to be taken seriously; readers will only consider the paper significant if it makes sense to them.

I think this analysis is consistent with the ability to remove 75% of DHSs (and the ability, *in vivo*, to remove 90% of the MCMs) without disrupting replication timing, as long as the removal of DHSs uniformly affects early and late replicating regions. If the DHSs were specifically removed from the late replicating regions, these late regions would only be replicated passively, and then very late; removal of the DHSs from early regions would make the affected early regions replicate later. (Moreover, I suspect the variation in the length of S will increase as DHSs are uniformly deleted, just as genome instability increases as MCM number is reduced, *in vivo*.)

Although the model is described well at the conceptual level, no technical details of the model or its implementation are provided. At least a brief description of how the model was implemented, both algorithmically and in terms of the software environments used, should be included in the

methods. Moreover, the model code should be made available. On a similar note, there is not indication of how the model was run. It is my impression that explicitly simulating replication for the whole human genome is computationally expensive. Is that true? If not, why not? What hardware was the model run on and how long did it take?

It would seem to be reasonable to explicitly cite a number of conceptually similar whole-genome modeling papers that have addressed replication kinetics in smaller genomes. In particular, I am thinking of work from the Bechhoefer (which is cited, but only in passing), De Moura and Lygeros labs, but there may be other that should be included, as well.

Reviewer #3:

In the present work, Gindin et al. show in the context of a simple stochastic replication model that correlating DNA replication initiation firing rates with the density of DNase hypersensitive sites. This an interesting idea, and I am sympathetic to the type of reasoning and modeling used. However, the specific case of DNase HS sites has already been considered previously (and dismissed as a correlation riding on 3D structural data). The authors will need to address and respond to this uncited work. Moreover, the simulation methods used are "clunky" and do not reflect the state of the art. At the very least, the authors should (a) be aware of what is possible with existing methods and (b) briefly mention / reference those.

Let us begin with the main conclusions of the paper. The authors assign a replication firing rate proportional to the ENCODE amplitude of various genomic features and find that the signal corresponding to DNase hypersensitive sites leads to a firing rate that, in a constant-rate approximation, is relatively consistent with the observed timing data. Moreover, the authors explore a host of less-successful correlates. So far, so good. However:

1) Ryba et al. (Genome Res. 20:761-770, 2010 -- not referenced in the present paper) show a very strong correlation between the Hi-C interaction map and timing profiles. I was surprised that this correlation is dismissed "in a reductionist spirit". I am sympathetic to the reductionist approach, but I do not see why the authors' correlate (DNase hypersensitive sites) is simpler than the 3D interaction map.

2) Even more relevant to the current paper, a recent paper by Baker et al. (PLoS Comp Biol 8(4), e1002443, 2012, again not referenced) also concludes that the Hi-C map controls replication timing, using different methods. It gives a more nuanced exploration of timing data (differentiating between types of domains in the genome, for example). Moreover, the authors explicitly consider the role of DNase I hypersensitive sites (using the same ENCODE data) and conclude that those sites play an incidental role, and the causal determinant is 3d structure. I'm all for the authors here if they want to make a counter-case that reverses the causal-dependent roles, but they have not made that case in the present manuscript. (Hint: can the approach of Baker et al. explain the various plasticity data that are discussed here?)

3) The idea that the number of simultaneously active forks may be rate limiting has been explored previously in a few different contexts. These should be discussed. See the discussion in Yang et al. Phys. Rev. E78, 041917, 2008. Also, Goldar et al. (PLoS ONE 4(6) e5899, 2009) build a model connecting initiation rates (again with an idea to explain time dependence).

4) I was surprised at the authors' figure for replication fork speeds (50 bases / sec.) They give no reference for this figure, and it is higher than what other studies report. For example, Guilbaud et al. (PLoS Comp. Biol. 7, e1002322, 2011) find rates in similar cell types that range from 0.7-2.0 kb/min (= 12-33 bps). These rates are consistent with a number of studies in other mammalian organisms. Adopting ceteris paribus a more conventional fork velocity would alter significantly the authors' conclusions regarding the agreement between their in silico S phase durations and observed in vivo S phase durations.

5) The firing rate (IPLS) has been assumed to be constant in time during S phase. This assumption should be stated explicitly and discussed. The context is that

(i) in studies of embryonic cells that average over genome positions, clear evidence for time-dependent rates has emerged, and, as discussed by Yang et al., *Physical Review E* 78, 041917 (2008), such rates play an important role in determining the width of the S-phase timing distribution (Fig. 3E here). One review (which argues that the time-dependent rates may have a universal form) is A. Goldar et al., *PLoS ONE* 4(6) e5899 (2009).

(ii) On the other hand, recent work by A. Demczuk et al. (*PLoS Biology* 10(7) e1001360, 2012) studied replication timing profiles in a particular genomic region in detail and concluded that local timing profiles (such as the ones looked at in the present paper) are not very sensitive to time variations. This would bolster the authors' case for neglecting such variations if their main concern is to match average timing profiles (which are not sensitive to some aspects of replication timing, as we now can understand).

(iii) Yang et al., *Physical Review E* 78, 041917 (2008) showed, via a modeling approach that the time dependence of the firing rate for replication in *Xenopus* embryos is close to (but not identical) to the function that would be predicted that minimizes the maximum number of simultaneously active replication forks (the parameter "N" in this paper). Again, this possible connection might be seen as reinforcing the significance of N (as advocated in the present work).

(iv) Adopting a firing rate that increases through (most of) S phase may reconcile the use of a slower fork velocity with the observed duration of S phase.

My other general set of concerns has to do with methodology:

1) There are far more efficient numerical algorithms outline that will speed up the code outlined here by several orders of magnitude. Once an origin has initiated there is no need to propagate a fork step by step. Rather, collisions can be computed directly. See, for ex., Fig. 5 of Jun et al., *Phys. Rev. E* 71, 011908 (2005).

2) Some jargon is needlessly obscure: The authors should define precisely what they mean by "IPLS" = initiation probability landscape". As I understand it, it is simply the firing rate $I(x,t)$ (sometimes called the initiation rate), defined as the number of origins initiated per time per length of unreplicated DNA. Here the rate is assumed constant, $I(x)$. (The fact that the simulation here allows initiation only if the DNA has not replicated leads to the conditioning on unreplicated DNA.) This quantity has been discussed extensively, for example in the "nomenclature" section at the end of Hyrien & Goldar (*Chrom Res.* 2010). Note that the present paper omits A. Goldar from the list of authors, an oversight that should be corrected. [Small aside: the authors should state more clearly that the local DNA replication time is an output of their model, not an input. I misread that the first time and had to go over the paper a couple of times before I understood what the authors did.]

3) Analytic results are available that have not been used. Given an "IPLS" or firing rate, one can directly compute the timing profile. See, for ex., R. Retkute et al., *Phys. Rev. E* 86, 031916, 2012. Using analytic results can bypass the need for extensive simulation (even using the more efficient algorithms described above). As an example of what one could get out of analytic results, Fig. 3E gives a histogram of the distribution of S-phase lengths. The work by Bechhoefer et al. (*Phys. Rev. Lett.* 98, 098105, 2007; *Phys. Rev. E* 78, 041917, 2008) argued that such distributions should be Gumbel for a wide range of models. The authors should check this, not only to test that prediction but also because a Gumbel distribution, having only 2 parameters (location and scale) is a convenient way to summarize the results of a model or simulation. By eye, a Gumbel form should give a decent fit.

In conclusion, while I am sympathetic to the general approach that the authors have taken, there are serious issues both with regard to specific results on DNase HS sites previously obtained (but not discussed here) and with regard to the general methodologies adopted. I think it quite possible that these issues could be addressed; however, the authors need to better digest the work that has previously been done.

Thank you for the opportunity to revise our paper. We have commented below on each of the points raised by the referees.

Reviewer 1

In this paper, the authors used Monte Carlo simulations to predict replication timing only 2 parameters: IPLS (initiation probability landscape) and N, the number of simultaneously active replication forks. IPLS is adopted from various genomic and epigenomic features, which basically determines the location of replication initiation. By using this prediction model, they show that one single feature - DNase hypersensitivity - can completely account for the timing profile. Other epigenetic marks correlate with RT due to their co-variation with DNase HS sites. The remarkable robustness of the model to random removal of 3/4 of DHSs is exactly what is expected of a replication system, given what we know of replication regulation in mammalian cells, and provides additional confidence in this model. The fact that it is independent of probability at each site challenges one of the major models of replication timing regulation; these data suggest it is the locations of sites not so much their efficiency. While readers may not be prepared to conclude that every DHS is an origin or vice versa, and I suspect many replication people will be resistant to this paper, the point here is that the simple principles of this model can account for the patterns of replication seen. This is a provocative and important study and should be published. There is just one point that I feel needs to be addressed and for which I would like to see the response.

1. *The discussion of the Besnard origins is very confusing: “A similar picture emerged when integrating experimental replication initiation data (Besnard et al, 2012), where initiation activity is strongly diminished at non-DNase HS overlapping marks when compared to the full dataset (Figure S6). Do the origins look the same or better or worse than DNaseI alone? If you take the origins and subtract DNaseI, what does that model look like? How do you interpret these results (depending on what they are)?*

We agree with the reviewer that the discussion would benefit from clarification. We expanded this section in the manuscript and we address it here as well.

Besnard and colleagues (Besnard *et al*, 2012) used deep-sequencing to map replication origins in human cells. In our analysis, we used the locations of these empirically determined origins and asked whether they overlap with genome features that, when fed into our model, predict DNA replication timing slightly less accurately compared to locations of DNaseI HS sites.

Looking at Figure S6, signal density is plotted as a function of genome distance to replication initiation site (centered around the 0 mark on the x axis). The black lines show that there is an enrichment, as measured by sequencing tag density, of JunD binding, H3k4me2, and H3k9ac histone marks at replication initiation sites. Crucially, this enrichment dissipates when DNaseI sites are removed from each of the datasets (red line). This experiment clearly shows that JunD binding, H3k4me2 and H3k9ac marks are only associated with DNA replication initiation sites when they overlap DNaseI sites.

We do not make an attempt to construct an IPLS from initiation sites mapped by Besnard and colleagues. Such a model would, at most, be able to mimic replication timing profiles. It would not, however, address the central reason for our work, which concerns itself with determining the genome factors that *determine* replication timing profiles.

2. *Some of the references are not correct in the second sentence of the introduction:*

- (a) *It is intimately associated with key aspects of cell biology, including cell differentiation (Hansen et al, 2010; Ryba et al, 2011b), - the first paper to show replication timing changes during differentiation at all is Hiratani et. al., PNAS, 2004, then the first to show it by genome-wide methods is Hiratani et. al., PloS Biology 2008. cancer progression (Donley & Thayer, 2013; Fritz et al, 2012), - this is where Ryba et. al. Genome Research 2012 should be referenced the 3D*

conformation of cellular DNA (Ryba et al, 2012; Moindrot et al, 2012) - the first paper to show this is Ryba et. al. Genome Research 2010.

We appreciate your pointing this out, we made the appropriate changes to the manuscript.

3. Some statements should be toned down:

(a) Bottom of first intro page: "robustly determined" - I suggest changing these phrases to: "can be accounted for"

Done

(b) Middle of pdf page 6 - same thing "main independent determinant".

We now condition our conclusion on "available data."

4. The correlations to cell types using cell type specific DHS data look good, but I did not see how the DNaseI model for one cell type matches or deviates from the replication timing of a different cell type. A simple correlation matrix might be nice to show.

(a) **We agree that this information is of benefit. Towards that end we include such a representation in Figure 3A, where we compare DNaseI model across three cell lines**

5. Figure 3C - this figure is not entirely clear. First, it is confusing because if this is an *in silico* translocation, and if DNaseI follows replication timing, shouldn't this be obvious. Clearly they do not have DHSs in the translocated cell line but, if the authors wish to compare their data to actual data, at a minimum they should use the replication timing profiles from a translocated cell line, which are published and available.

(a) **Figure 3C shows that abrupt changes in replication timing, which have been observed around chromosome fusions sites, are the consequence of mapping rearranged chromosomal data onto the normal genome and that this phenomenon is influenced by the density of DNase HS sites on either side of the breakpoint. Ideally, we would perform the simulation using DNase HS data from REH cells, which harbor the translocation, and compare it to the empirical REH DNA replication timing data. However, REH DNase HS data are not readily available.**

We agree with the reviewer that the readers would benefit from a presentation where *in-silico*-generated GM06990 replication timing data are compared to observed replication timing profiles of REH cells. To that end, we now provide such a comparison in Figure S9. Examining Figure S9, both *in-silico*-generated GM06990 replication timing data and REH experimentally determined data show an abrupt change from early replication timing to late replication timing around the site of the breakpoint.

6. There was a paper by Eric Bouhassira in Genome Research, 2009 where he predicted replication timing from the promoters of transcribed genes. How do the authors' data differ? Clearly the match here is better - can the author's provide a satisfactory explanation.

(a) **Desprat et al. (Desprat et al, 2009) used regression analysis to derive a relationship between replication timing and distance to highly expressed (top quartile) genes in the form of $1/(ax + b)$, where x is genomic distance and a and b are constants that were adjusted by trial and error. The major difference between Desprat et al. and the work described here is that our approach does not depend on prior knowledge of DNA replication timing for predictions. Secondly, in this work we are able to show that the patterns of DNA replication timing arise from the uncoordinated firing of well-defined DNA replication initiation sites that are localized by sites of DNase hypersensitivity. These observations require, in the absence of empirical data, an accurate model of DNA replication, which is provided here.**

7. Figure 1B was not clear to me at all. What are the lines? Can the authors explain a bit more. Also, it seems that since it is showing a correlation, that some kind of R value should be shown in the figure, if I understand it correctly?
 - (a) **Figure 1B is a contour plot, we've added additional details to the figure caption to clarify. We've also added the correlation value to the figure.**
8. In Figure 3A, it would help to use correlation values in addition to the distances in the dendrogram to see the differences better.
 - (a) **We've added a heat-map beneath the dendrogram to visualize the correlations**

Reviewer 2

Girdin et al. describe an elegantly simple computational model for human DNA replication and use it to explore the factors that determine replication timing. They come to the surprising, but apparently robust, conclusion that timing is determined almost exclusively by the location of origins and not any intrinsic timing or efficiency characteristic of the origins. Moreover, they show that the position of origins is best predicted by DNase I hypersensitive sites (DHSs) and that other genome features that correlate with origins do so primarily because they too correlate with DHSs. These results have important mechanistic implications for biochemical models of how replication is regulated. As such, it will be of interest to a broad audience interested in DNA replication and genome stability. Nonetheless, the following issues need to be addressed before publication.

1. The conclusion that only the location of origins is important, and that the timing or efficiency of individual origin firing is irrelevant, is initially counterintuitive. However, after reading the discussion very carefully, I think I understand what is going on. It seems to me that the key is the going to be the density of origins/DHSs, with more origins/kb specifying early regions and less origins/kb specifying late regions. If this interpretation is correct, the authors should depict origin density v. timing, perhaps as heat maps, to make this point clear. If not, they should explain, at least to me, why this interpretation is incorrect. It seems to me that this interpretation is conceptually similar to the Multiple Initiator Model of Yang et al. (<http://www.ncbi.nlm.nih.gov/pubmed/20739926>), which the authors reference, in which the behavior of any one initiator (MCMs for Yang, DHSs for Girdin) is uniform and inconsequential; what matters is the number of initiator (at an origin for Yang or in a region of initiation for Girdin). A key factor in this interpretation is the difference in resolution of DHS maps (1kb) and the replication profiles (100kb) (and here I am referring to the resolution of the experimental replication profiles and the output of the model (i.e., Figure 1C), not the 500 bp resolution at which the model is calculated), which allows many DHS sites in every region of replication initiation, even in relatively DHS-poor late replicating regions. However, it took me a lot of work to make this connection. This analysis is implicit in the description and discussion of the paper's results, but I think it would be a great service to many readers, who may not make the required conceptual connections, if it was spelled out much more explicitly. It will also make the paper more likely to be taken seriously; readers will only consider the paper significant if it makes sense to them.
 - (a) **These are excellent observations and we appreciate helpful suggestions to improve the manuscript. We've significantly expanded the Discussion section to cover many of the points addressed by the above remark. We've added a representation of DNase HS site density to Figure 1C. With regard to specific points, we illustrate in Figure S14 that, by far, most of DHS sites reside in early replication regions. We also attempted to correlate DHS density directly to replication timing using different window sizes (Figure S13), drawing a conclusion that DNA replication timing is determined by *both* the local density of DNase HS sites and the global mechanism of replication fork movement .**
2. I think this analysis is consistent with the ability to remove 75% of DHSs (and the ability, in vivo, to remove 90% of the MCMs) without disrupting replication timing, as long as the removal of DHSs

uniformly affects early and late replicating regions. If the DHSs were specifically removed from the late replicating regions, these late regions would only be replicated passively, and then very late; removal of the DHSs from early regions would make the affected early regions replicate later. (Moreover, I suspect the variation in the length of S will increase as DHSs are uniformly deleted, just as genome instability increases as MCM number is reduced, *in vivo*.)

- (a) **This is an interesting hypothesis compatible with our model. We expect that the timing in early replicating regions would be only moderately affected, mostly on the boundary of these regions, stretching out the transition region between early and late replicating.**
3. *Although the model is described well at the conceptual level, no technical details of the model or its implementation are provided. At least a brief description of how the model was implemented, both algorithmically and in terms of the software environments used, should be included in the methods. Moreover, the model code should be made available. On a similar note, there is no indication of how the model was run. It is my impression that explicitly simulating replication for the whole human genome is computationally expensive. Is that true? If not, why not? What hardware was the model run on and how long did it take?*
- (a) **In response to the reviewer, we've added a paragraph to the Methods section describing practical aspects of the model. Additionally, we plan to submit a technical description of the model to a bioinformatics journal providing the executables, accompanying source-code, and the analysis tools that were used here. The model is not computationally expensive, it is written in highly efficient C++, and could be executed in a multi-threaded mode, decreasing the computational time even further. When running on four cores of a 2.93GHz Linux machine, it takes roughly 15 minutes to simulate replication timing for human chromosome 1 (the longest human chromosome). With access to a reasonable computational cluster, and since the model is chromosome-centric, the replication timing for all chromosomes could be calculated in parallel. It would therefore take around 15 minutes to simulate the replication timing program for all human chromosomes.**
4. *It would seem to be reasonable to explicitly cite a number of conceptually similar whole-genome modeling papers that have addressed replication kinetics in smaller genomes. In particular, I am thinking of work from the Bechhoefer (which is cited, but only in passing), De Moura and Lygeros labs, but there may be other that should be included, as well.*
- (a) **We have expanded the Introduction to include previous modeling efforts in yeast and amphibian embryos.**

Reviewer 3

In the present work, Gindin et al. show in the context of a simple stochastic replication model that correlating DNA replication initiation firing rates with the density of DNase hypersensitive sites. This is an interesting idea, and I am sympathetic to the type of reasoning and modeling used. However, the specific case of DNase HS sites has already been considered previously (and dismissed as a correlation riding on 3D structural data). The authors will need to address and respond to this uncited work.

Moreover, the simulation methods used are "clunky" and do not reflect the state of the art. At the very least, the authors should (a) be aware of what is possible with existing methods and (b) briefly mention / reference those.

We are aware of the mentioned algorithmic and analytical methods, yet none of them are applicable to our model (See detailed response below). We believe that this and subsequent questions stem from a misinterpretation of the "Initiation Probability Landscape" $\mathcal{I}(x, t)$ in our model as an "Initiation Rate" $I(x, t)$ as used in existing models. (See also the discussion further below)

Let us begin with the main conclusions of the paper. The authors assign a replication firing rate proportional to the ENCODE amplitude of various genomic features and find that the signal corresponding to DNase hypersensitive sites leads to a firing rate that, in a constant-rate approximation, is relatively consistent with the observed timing data. Moreover, the authors explore a host of less-successful correlates. So far, so good. However:

It seems important to emphasize that the predicted observable in our model is not the firing rate (as apparently implied by the referee) but the global replication timing program. We make no attempt to directly compare firing rates. Also, we discuss extensively that the “proportionality” between the ENCODE signal and initiation rates is not essential for the global timing program. In fact, this has been one of the main observations in the manuscript: it is the *location* and *not* the amplitude of the DNASE HS signal that control timing.

We would also like to point out that the time-independence of the IPLS $\mathcal{I}(x, t) = \mathcal{I}(x)$ is *not* an approximation. The referee is probably expecting that the (related but different) initiation rate $I(x, t)$ is time-dependent, as demonstrated experimentally and theoretically in a number of studies. In our model, time dependence of $I(x, t)$ is a *kinematic* result of our mechanistic model. We discuss this in more detail below.

1. Ryba *et al.* (*Genome Res.* 20:761-770, 2010 – not referenced in the present paper) show a very strong correlation between the Hi-C interaction map and timing profiles. I was surprised that this correlation is dismissed “in a reductionist spirit”. I am sympathetic to the reductionist approach, but I do not see why the authors’ correlate (DNase hypersensitive sites) is simpler than the 3D interaction map.

We did not intend to dismiss the strong (and unquestionable) correlation of the replication timing signal with the first eigenvector of the Hi-C contact matrix. We agree that the original sentence in the discussion could be interpreted in a way that we use the argument that the 3D conformation is more “complex”¹ to eliminate it as a viable competitor. This is not the case, we do not use the complexity argument but rather reject the first EV of the HI-C contact because the correlation of EV1 with timing is only $r = 0.8$ (Ryba *et al.*), *i.e.* if it were included in our comparison, EV1 would rank as the 34th best model among all tested models. In response to the reviewer’s remark we have re-phrased the corresponding discussion to clarify this point.

2. Even more relevant to the current paper, a recent paper by Baker *et al.* (*PLoS Comp Biol* 8(4), e1002443, 2012, again not referenced) also concludes that the Hi-C map controls replication timing, using different methods. It gives a more nuanced exploration of timing data (differentiating between types of domains in the genome, for example). Moreover, the authors explicitly consider the role of DNase I hypersensitive sites (using the same ENCODE data) and conclude that those sites play an incidental role, and the causal determinant is 3d structure. I’m all for the authors here if they want to make a counter-case that reverses the causal-dependent roles, but they have not made that case in the present manuscript. (Hint: can the approach of Baker *et al.* explain the various plasticity data that are discussed here?)

We have added a discussion regarding the observations that the genome organization may play a prominent role in DNA replication timing as suggested by the qualitative observation that the boundaries of replication domains (Baker *et al.*, 2012) coincide with the boundaries of the contact probability map of one cell line.

We, having read and discussed work by Baker and colleagues (Baker *et al.*, 2012), could not come to the conclusion that the authors of that manuscript categorically rule out the DNase HS sites as causal factors of DNA replication timing. Baker *et al.*, 2012 demonstrate that the end-points of U-shaped replication domains co-localize with open chromatin and further show a strong correlation between DNA replication timing and DNase HS. This

¹In our judgment, there are, actually, good reasons to consider DNASE HS sites as less complex. The formation of DNASE HS sites is reasonably well understood as a result of sequence dependent TF binding leading a displacement of histones (see e.g. Felsenfeld *et al.* (1996)). On the other hand, the 3d contact matrix is, by any measure, not well understood. It is not even clear if there is a single dominating geometry or if the contact matrix is a superposition of hundreds of thousands of geometries

does not suggest to us, nor is it explicitly mentioned in the manuscript itself, that DNase HS are somehow “incidental” to DNA replication timing.

3. The idea that the number of simultaneously active forks may be rate limiting has been explored previously in a few different contexts. These should be discussed. See the discussion in Yang *et al.* *Phys. Rev. E* 78, 041917, 2008. Also, Goldar *et al.* (*PLoS ONE* 4(6) e5899, 2009) build a model connecting initiation rates (again with an idea to explain time dependence).

Following the reviewers recommendation we have added a brief discussion of these papers to the manuscript .

4. I was surprised at the authors’ figure for replication fork speeds (50 bases / sec.) They give no reference for this figure, and it is higher than what other studies report. For example, Guilbaud *et al.* (*PLoS Comp. Biol.* 7, e1002322, 2011) find rates in similar cell types that range from 0.7-2.0 kb/min (= 12-33 bps). These rates are consistent with a number of studies in other mammalian organisms. Adopting *ceteris paribus* a more conventional fork velocity would alter significantly the authors’ conclusions regarding the agreement between their *in silico* S phase durations and observed *in vivo* S phase durations.

- (a) The speed of the eukaryotic replication fork is commonly reported as 50 bases/seconds (3 kb/min) (see for instance Alberts *et al.*, 2008). We agree that the speed of the replication fork is variable, with a common reference range being 0.5 to 5 kb/min (Kornberg and Baker, 1992), which is in general agreement that (1) eukaryotic forks are 10-fold slower than prokaryotic forks, and (2) eukaryotic fork velocities vary over a 10-fold range (Conti *et al.*, 2007; Hyrien and Goldar, 2010). While Guilbaud *et al.*, 2011 do indeed calculate replication fork velocities between 0.7 and 2.0 kb/min for HeLa cells, they also reference velocities that range from 1.73 to 2.9 kb/min in fibroblast cells and velocities that range from 2.06 and 4.4 kb/min in lymphoblastoid cells. Therefore, we disagree that a replication fork traveling at 3 kb/min assumed in the estimation of the length of the S-phase is outside the range of values.

5. The firing rate (IPLS) has been assumed to be constant in time during S phase. This assumption should be stated explicitly and discussed. The context is that

- (a) in studies of embryonic cells that average over genome positions, clear evidence for time-dependent rates has emerged, and, as discussed by Yang *et al.*, *Physical Review E* 78, 041917 (2008), such rates play an important role in determining the width of the S-phase timing distribution (Fig. 3E here). One review (which argues that the time-dependent rates may have a universal form) is A. Goldar *et al.*, *PLoS ONE* 4(6) e5899 (2009).

On the other hand, recent work by A. Demczuk *et al.* (*PLoS Biology* 10(7) e1001360, 2012) studied replication timing profiles in a particular genomic region in detail and concluded that local timing profiles (such as the ones looked at in the present paper) are not very sensitive to time variations. This would bolster the authors’ case for neglecting such variations if their main concern is to match average timing profiles (which are not sensitive to some aspects of replication timing, as we now can understand).

Following the reviewers suggestion, we now state more forcefully that the IPLS is constant in time, by design. We also include a discussion of how the (time-constant) IPLS $\mathcal{I}(x)$ and the time-dependent initiation rate $I(x,t)$ are related, showing that the universal time-dependence of I can be qualitatively understood as a kinematic consequence of the proposed mechanistic process, see supplemental figure S15. (We believe that a quantitative discussion about if and how well our model re-creates the universal $I(t)$ behavior is beyond the scope of the current paper and we plan on submitting a separate paper on that subject). Essentially, $I(x,t)$ is the product of the ‘diffusion’ process (selecting location x), the density of initiation sites and the probability that x is already replicated at t . Measuring $I(t)$ numerically shows (Supplementary Figure S15) the expected increase in the first half of the S-phase

and a decrease later on. In our case, a lower density of initiation sites (i.e. DNASE HS sites) in the late replicating region leads to this drop-off through a longer 'diffusion' time to find an unreplicated initiator. Therefore, we did not have to utilize a more complex diffusion model such as the anomalous diffusion used by Gauthier and Bechhoefer (Gauthier and Bechhoefer, 2009) in *Xenopus*.

- (b) Yang *et al.*, *Physical Review E* 78, 041917 (2008) showed, via a modeling approach that the time dependence of the firing rate for replication in *Xenopus* embryos is close to (but not identical) to the function that would be predicted that minimizes the maximum number of simultaneously active replication forks (the parameter "N" in this paper). Again, this possible connection might be seen as reinforcing the significance of N (as advocated in the present work).

We agree with the reviewer. While we have not performed a formal analysis, we expect that the described difference between the number of *active* replication forks and the time dependence can be accounted for by observing that the total time dependence function is a product of the number of *available* forks and the diffusion process to find initiation sites.

6. My other general set of concerns has to do with methodology:

- (a) There are far more efficient numerical algorithms outline that will speed up the code outlined here by several orders of magnitude. Once an origin has initiated there is no need to propagate a fork step by step. Rather, collisions can be computed directly. See, for ex., Fig. 5 of Jun *et al.*, *Phys. Rev. E* 71, 011908 (2005).

The speed-up in the "Double-list algorithm" in that paper is largely achieved by first estimating the number N of initiation events by using the Poisson distribution taking into account the number of available (un-replicated) potential initiation sites and then subsequently randomly choosing only N initiation sites. The 100-1000 fold speedup is in comparison to their naive (lattice) algorithm, where each of the L potential initiation sites is visited and initiated with a certain probability (leading to an algorithm scaling as $O(L)$). In our model, this speed-up is neither necessary nor possible, because initiation is not associated with DNA, but with the rate limiting factors. The performance of the algorithm without further optimizations is therefore not $O(L)$ but $O(F)$, where the number F of rate limiting factors, which is several orders of magnitude smaller than the length L of the simulated genome. We do also keep track of un-engaged rate limiting factors, utilizing a double-linked list implementation used earlier by us (Bilke *et al* (1995)) and others, so that throughout most parts of the simulated S-phase, when almost all rate limiting factors engaged, the performance of the algorithm concerned with initiation site selection is $O(1)$.

The phantom-nuclei algorithm does not improve the performance of our model because \mathcal{I} is time-independent in our model *and* the number of potential initiation sites is very large, a situation where, according to Jun *et al* (Jun *et al* (2005)), this algorithm performs poorly.

- (b) Some jargon is needlessly obscure: The authors should define precisely what they mean by "IPLS" = initiation probability landscape". As I understand it, it is simply the firing rate $I(x,t)$ (sometimes called the initiation rate), defined as the number of origins initiated per time per length of un-replicated DNA. Here the rate is assumed constant, $I(x)$. (The fact that the simulation here allows initiation only if the DNA has not replicated leads to the conditioning on un-replicated DNA.) This quantity has been discussed extensively, for example in the "nomenclature" section at the end of Hyrien & Goldar (*Chrom Res.* 2010). Note that the present paper omits A. Goldar from the list of authors, an oversight that should be corrected. [Small aside: the authors should state more clearly that the local DNA replication time is an output of their model, not an input. I misread that the first time and had to go over the paper a couple of times before I understood what the authors did.]

We hope to have clarified that $\mathcal{I}(x,t) \neq I(x,t)$.

Figure 1: Gumbel distribution (red line) plotted over the distribution of S phase lengths (dark bars). Gumbel distribution location parameter (μ) = 5340.8; scale parameter (σ) = 877.1

- (c) *Analytic results are available that have not been used. Given an "IPLS" or firing rate, one can directly compute the timing profile. See, for ex., R. Retkute et al., Phys. Rev. E 86, 031916, 2012. Using analytic results can bypass the need for extensive simulation (even using the more efficient algorithms described above). As an example of what one could get out of analytic results, Fig. 3E gives a histogram of the distribution of S-phase lengths.*

The referee correctly points out that the mentioned analytical results were derived for $I(x, t)$. Unfortunately, there is no straight-forward way to adopt these equations to our model based on $\mathcal{I}(x, t)$. While it is possible to derive an expression relating the two and also for the number of available rate limiting factors, one is ultimately led to evaluate the non-local integrals Kolmogorov's argument was designed to avoid, unless a spatially homogeneous IPLS is used, as in Gauthier&Bechhoefer 2009 (Gauthier and Bechhoefer (2009)). Also, unless one assumes sufficiently "friendly" initiation probabilities, extensive numerical integration is still needed to evaluate the analytical expressions.

- (d) *The work by Bechhoefer et al. (Phys. Rev. Lett. 98, 098105, 2007; Phys. Rev. E78, 041917, 2008) argued that such distributions should be Gumbel for a wide range of models. The authors should check this, not only to test that prediction but also because a Gumbel distribution, having only 2 parameters (location and scale) is a convenient way to summarize the results of a model or simulation. By eye, a Gumbel form should give a decent fit.*

That's a great observation, see Figure 1 in this document.

In conclusion, while I am sympathetic to the general approach that the authors have taken, there are serious

issues both with regard to specific results on DNase HS sites previously obtained (but not discussed here) and with regard to the general methodologies adopted. I think it quite possible that these issues could be addressed; however, the authors need to better digest the work that has previously been done.

We would like to thank the reviewers again for their critical remarks and hope that we replied to all of their concerns.

References

- Alberts B, Wilson JH, Hunt T (2008) *Molecular Biology of the Cell*. Garland Science, 5th edition
- Baker A, Audit B, Chen CL, Moindrot B, Leleu A, Guilbaud G, Rappailles A, Vaillant C, Goldar A, Mongelard F, D'Aubenton-Carafa Y, Hyrien O, Thermes C, Arneodo A (2012) Replication fork polarity gradients revealed by megabase-sized U-shaped replication timing domains in human cell lines. *PLoS Comput Biol* **8**: e1002443
- Besnard E, Babled A, Lapasset L, Milhavet O, Parrinello H, Dantec C, Marin JM, Lemaitre JM (2012) Unraveling cell type-specific and reprogrammable human replication origin signatures associated with G-quadruplex consensus motifs. *Nat Struct Mol Biol* **19**: 837–44
- Bilke S, Burda Z, Jurkiewicz J (1995) Simplicial Quantum Gravity on a Computer. *Comput Phys Commun* **85**: 278–292
- Conti C, Saccà B, Herrick J, Lalou C, Pommier Y, Bensimon A (2007) Replication fork velocities at adjacent replication origins are coordinately modified during DNA replication in human cells. *Mol Biol Cell* **18**: 3059–67
- Desprat R, Thierry-Mieg D, Lailier N, Lajugie J, Schildkraut C, Thierry-Mieg J, Bouhassira EE (2009) Predictable dynamic program of timing of DNA replication in human cells. *Genome Res* **19**: 2288–99
- Felsenfeld G, Boyes J, Chung JAY, Clark D, Studitsky V (1996) Chromatin structure and gene expression. *Proc Natl Acad Sci* **93**: 9384–9388
- Gauthier M, Bechhoefer J (2009) Control of DNA Replication by Anomalous Reaction-Diffusion Kinetics. *Phys Rev Lett* **102**: 158104
- Guilbaud G, Rappailles A, Baker A, Chen CL, Arneodo A, Goldar A, D'Aubenton-Carafa Y, Thermes C, Audit B, Hyrien O (2011) Evidence for sequential and increasing activation of replication origins along replication timing gradients in the human genome. *PLoS Comput Biol* **7**: e1002322
- Hyrien O, Goldar A (2010) Mathematical modelling of eukaryotic DNA replication. *Chromosome Res* **18**: 147–61
- Jun S, Zhang H, Bechhoefer J (2005) Nucleation and growth in one dimension. I. The generalized Kolmogorov-Johnson-Mehl-Avrami model. *Phys Rev E* **71**: 011908
- Kornberg A, Baker TA (1992) *DNA Replication*. University Science

2nd Editorial Decision

29 January 2014

Thank you again for sending us your revised manuscript. As you will see below, both referees are satisfied with the modifications made and are now positive on your study.

With regard to the modeling work described in this study, we consider that the computational model represents a central and integral part of the work. In view of our policy on availability of materials, data and software (<http://msb.embopress.org/authorguide#a2.4>), we would therefore kindly ask you to include the model in the manuscript in a machine-readable form (as zipped dataset) and to include an appropriate description and documentation so that others can re-use the model, reproduce your analysis and build upon your work.

On a more editorial level, we would like to ask you to provide individual files for all Supplementary Figures. Each file should contain the Supplementary Figure and the corresponding Supplementary Figure legend. Please make sure that you use the whole figure for each file and not figure panels (i.e. the current Figures S1A and S1B have to be provided as Figure S1).

Please note that you can also supply 'source data' for specific figures displaying important data. In the context of the present study, figures comparing simulation data to empirical data seem to be particularly relevant. See further instructions at <http://msb.embopress.org/authorguide#a3.4.3>.

Please resubmit your revised manuscript online, with a covering letter listing amendments and responses to each point raised by the referees. Please resubmit the paper ****within one month**** and ideally as soon as possible. If we do not receive the revised manuscript within this time period, the file might be closed and any subsequent resubmission would be treated as a new manuscript. Please use the Manuscript Number (above) in all correspondence.

REFeree REPORTS

Reviewer #2:

The authors have addressed all of my concerns. I think it would be useful for them to explicitly state that Replicon and other software written for this paper will be made available via publication elsewhere.

Reviewer #3:

The authors have done a good job in responding to the points raised by me and by the other reviewers. The new version is clearer than the original. The authors make an interesting, somewhat unexpected hypothesis, support it reasonably with simulations and data, and make a reasonable effort to put their work in context with previous studies. It will be an interesting contribution to the literature.

2nd Revision - authors' response

30 January 2014

Thank you for considering the above referenced revised manuscript entitled "A chromatin structure based model accurately predicts DNA replication timing in human cells." We appreciate the positive comments of the reviewers. We have made the requested editorial changes and have added the executable software and accompanying data files to the resubmission. Thank you for your attention to our revision.

3rd Editorial Decision

05 February 2014

Thank you for adding the executable binaries of your Replicon model. We appreciate that you included a short README file to provide some information regarding the use of the program. This would however not be sufficient for others to understand, reproduce and build upon your computational work.

In this study, you "built a minimal model", benchmarked it and used it with a variety of initiation probability landscapes (IPLS) to infer the determinants of replication timing. We thus consider that Replicon is a *central* and *integral* component of this study and that appropriate description of the software and its use is mandatory in this case.

For publication, we would therefore kindly ask you to provide all the necessary information, according to our policy regarding the availability of data and software published in Molecular Systems Biology (<http://msb.embopress.org/authorguide#a2.4>).

Before we can formally accept your study for publication, the following points need to be addressed:

- We are pleased to see that the README file points to an open source Google Code page, but the corresponding code repository was empty at the time of writing this letter. For long-term archival we would ask you to provide the code as part of the supplementary information.
- Please provide a description of all the options that can be used when running Replicon.
- We appreciate that the current README file includes a reference to the database table and scheme used for the HS sites, but a complete description of how various IPLS files can be generated and used as input to Replicon should be provided.
- Please provide the files corresponding to the IPLS used in the analysis.

3rd Revision - authors' response

11 February 2014

Thank you for considering the above referenced revised manuscript entitled "A chromatin structure based model accurately predicts DNA replication timing in human cells." In this revision, we have provided the additional software and IPLS information as you requested. This includes the full source code for the model software (Replicon). We have also added a README file with very detailed instructions on how to reproduce our findings and use our application on new data (extending our work). Note that we have also included the IPLS for GM06990 and the timing predictions generated from this IPLS. Due to the large file size of the IPLS for each sample and chromatin mark, rather than upload all of these we have provided clear instructions for the simple process of converting any set of genome annotations to the IPLS format compatible with our software in the README document. Thank you for your attention to our revised manuscript package.

Acceptance letter

12 February 2014

Thank you for providing the source code and the instructions for the model software. I am pleased to inform you that your paper has been accepted for publication.